# EigenLoRAx: Efficient Low Rank Adaptation Using Recycled Principal Subspaces

## Abstract

The rapid growth of large models has raised concerns about their environmental impact and equity in accessibility due to significant computational costs. Low-Rank Adapters (LoRA) offer a lightweight solution for finetuning large models, resulting in an abundance of publicly available adapters tailored to diverse domains. We ask: Can these pretrained adapters be leveraged to further streamline adaptation to new tasks while addressing these challenges? We introduce EigenLoRAx, a parameter-efficient finetuning method that recycles existing adapters to create a principal subspace aligned with their shared domain knowledge which can be further augmented with orthogonal basis vectors in low-resource scenarios. This enables rapid adaptation to new tasks by learning only lightweight coefficients on the principal components of the subspace, eliminating the need to finetune entire adapters. EigenLoRAx requires significantly fewer parameters and memory, improving efficiency for both training and inference. Our method demonstrates strong performance across diverse domains and tasks, offering a scalable solution for edge-based applications and equitable deployment of large models in resource-constrained environments.

## 1 Introduction

Recent advancements in machine learning have driven the rise of large-scale models with billions of parameters. However, the size and complexity of these models not only make it impractical for most researchers to train or fine-tune them on downstream tasks but also contribute significantly to their carbon footprint, raising concerns about environmental sustainability. To address these challenges, there has been growing interest in parameter-efficient finetuning (PEFT) methods, such as adapters (Houlsby et al., 2019; Chen et al., 2022; Luo et al., 2023), low rank adaptation (LoRA) methods (Hu et al., 2021; Kopiczko et al., 2023; Liu et al., 2024), prompt-based methods (Lester et al., 2021; Razdaibiedina et al., 2023; Fischer et al., 2024). LoRA and its follow-up works (Meng et al., 2024; Liu et al., 2024) have gained significant attention for their simplicity. This has led to the proliferation of thousands of low-rank adapters within the growing open-source community. Given that these adapters are underutilized, an important question arises: Can we recycle the information contained in them to improve the efficiency of subsequent tasks? Recent work has shown that weight updates in deep neural networks occur within low-dimensional invariant subspaces (Kwon et al., 2024), aligning with the universality hypothesis that neural network behavior and learned representations often reside in shared and structured subspaces Chughtai et al. (2023); Guth & Ménard (2024). This suggests that LoRA adapters may similarly share a reusable *principal subspace* , eliminating the need to rediscover it during the training of new adapters.

We introduce **EigenLoRAx**, a parameter-efficient fine-tuning (PEFT) method that leverages this insight by decomposing the weights of a set of trained adapters into principal components, identifying a compact, information-dense subspace. EigenLoRAx reduces the number of learnable parameters by up to $100\times$ compared to LoRA, accelerates optimization by up to $2\times$ for new adapters, and enables more memory-efficient inference with multiple task adapters, particularly benefiting edge devices (Liu et al., 2022). Additionally, in low-resource domains, we demonstrate that EigenLoRAx can be further enhanced by augmenting the principal subspace with random components, orthogonalized with respect to the existing subspace, preserving its efficiency and performance.

Furthermore, we provide an initial theoretical analysis of EigenLoRAx. Our experiments across a wide range of vision and language tasks demonstrate its versatility and effectiveness, reinforcing the potential of shared subspaces in neural network adaptation.

Figure 1 provides an overview of our method. We introduce **EigenLoRAx**, which recycles pretrained adapters by identifying a shared *task-invariant* weight subspace. We hypothesize (and validate experimentally) that task-specific weights lie within this subspace, allowing for more efficient training with fewer parameters. This reduces memory footprint and enhances inference efficiency by enabling simultaneous serving of multiple adapters. EigenLoRAx is among the first to recycle pretrained adapters, replacing many while improving further training efficiency. Our key contributions are:

- (**Training**): EigenLoRAx uses up to $100\times$ **fewer parameters than LoRA** and converges up to $2\times$ **faster** than comparable methods with similar or better performance.

- (**Inference**): EigenLoRAx enhances **memory efficiency during inference** by approximately $18\times$ on multiple tasks, reducing the number of switchable parameters between tasks.

- (**Applicability**): We demonstrate the effectiveness of EigenLoRAx across a wide range , including text and image data, validating the existence of shared principal subspaces across modalities. It also retains performance in **zero-shot** and **low resource** scenarios.

- (**Scaling**): EigenLoRAx can be scaled up to recycle hundreds of underutilized adapters.

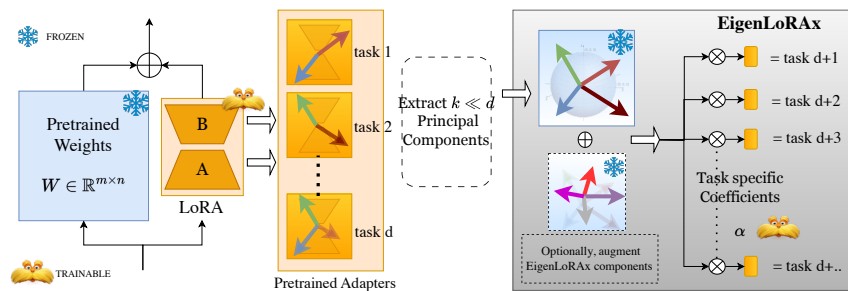

Figure 1: LoRA uses low-rank matrices for task-specific finetuning. We observe that LoRA adapters share a principal subspace across task domains. By recycling pretrained adapters, we extract *task-invariant* principal components, enabling efficient representation of both existing and future LoRAs using compact *task-specific* coefficients. This improves training speed, parameter efficiency, and memory usage. In low-resource settings, where adapters are scarce, we augment the subspace with randomly initialized components, ensuring orthogonality via the Gram-Schmidt process, ensuring they complement the extracted subspace without redundancy.

## 2 RELATED WORK

Low-Rank Adaptation (LoRA) models weight updates using low-rank matrices instead of full-weight training, a direction rooted in Burer-Monteiro factorization (Burer & Monteiro, 2003). LoRA (Hu et al., 2021) revived this idea for LLM finetuning, with variants emerging across domains (Ma et al., 2024; Chi et al., 2019; Kwon et al., 2024). However, with growing model sizes, even low-rank methods become expensive—for example, LoRA with rank 16 on GPT-3 (Brown et al., 2020) requires 75.5M parameters.

To improve efficiency, mixture-of-experts models (Huang et al., 2023; Wu et al., 2024; Diao et al., 2023; Zhong et al., 2024; Zhou et al., 2018) combine multiple low-rank modules. However, these require numerous high-quality adapters (Ku et al., 2024) and incur memory overhead (Zhou et al., 2022), along with instability from complex gating mechanisms (Zoph et al., 2022).

Recent work focuses on improving subspace initialization. Meng et al. (2024) show singular-vector-based LoRA initialization outperforms random, and Sharma et al. (2023) suggest discarding minor singular components for robustness. Other approaches use random principal components (Kopiczko et al., 2023) or weight matrices (Koohpayegani et al., 2024) to reduce parameter count, but often suffer from poor task alignment, as shown in Section 4. In contrast, EigenLoRAx extracts a *task-aligned principal subspace* from trained adapters, enabling better initialization and improved parameter

efficiency. Although our method primarily relies on LoRA (Hu et al., 2021), it can be applied to other PEFT methods (Liu et al., 2024; Zhang et al., 2023) by analyzing shared task-specific weights.

## 3 METHOD

In this section, we present the theoretical foundation Section 3.1 and algorithmic details Section 3.2 of our method, followed by a discussion on hyperparameter selection and an assessment of its practical advantages. The terms EigenLoRA, EigenLoRAx, ELoRA and ELoRAx are used interchangeably.

### 3.1 THEORETICAL PRELIMINARIES

For a full rank weight matrix $W \in \mathbb{R}^{m \times n}$ that learns to map input space $X \in \mathbb{R}^m$ to output space $\mathbb{R}^n$, the rank is expressed as $\min(m, n)$. As the rank of $W$ increases, modifying it to accommodate new tasks becomes computationally expensive and increasingly complex. This is a common challenge faced when finetuning pretrained large foundation models. LoRA is a parameter efficient finetuning approach used for large pretrained models with weights $W_0$ that mitigates this challenge by merely learning low-rank weight updates $W$ such that the risk between $Y$ and $W_0 X + W X + b$ is minimized. Instead of directly learning $W$, LoRA proposes to learn a lower ranked decomposition of $W$ by learning two low-rank matrices, $B \in \mathbb{R}^{m \times r}$ and $A \in \mathbb{R}^{r \times n}$, both having ranks $r$. This factorization ensures that the product $BA$ retains the original dimensions of $W_0$ while having a significantly reduced rank. As a result, although the transformation defined by $BA$ maps from $\mathbb{R}^m$ to $\mathbb{R}^n$, it does not span the full space of such mappings due to its constrained rank. The low-rank weight matrices result in substantially smaller number of trainable parameters than the full rank parameter count of $m \cdot n$. Such parameter efficient finetuning makes LoRA a computationally viable alternative for fine-tuning large-scale models.

Previous works such as Meng et al. (2024); Liu et al. (2024) have proposed the existence of a common parameter subspace implying the idea of shared principal subspace. We highlight that LoRA adapters share such a lower dimensional shared principal subspace when finetuned for diverse tasks. Along with reduction in computational overhead, it reinforces the idea that task-relevant transformations reside within a compact, reusable subspace. To formalize this, we first define a space of tasks representable by linear transformation matrices, providing a foundation for analyzing the role of shared principal subspaces in model adaptation.

**Definition 3.1** (Task definition for LoRAs). We first define a LoRA task $t_i(X_i, Y_i) : \mathbb{R}^m \to \mathbb{R}^n$ such that $Y_i = W_i^* X_i + b$ where $b$ is some constant. Then the LoRA task domain $\mathcal{T}_d$ is a set of $d$ such tasks, $\mathcal{T}_d = \{t_i\}_{i=1}^d$.

For a given set of pretrained weights (such as those from a foundation model) $W_0 \in \mathbb{R}^{m \times n}$, LoRA weights $BA$ at any layer modify the output as $W_0 X + BAX + \epsilon_t$, allowing the model to adapt to the new task and converge toward the optimal solution $W_t^*$. The key point here is that only $B$ and $A$ weights are updated during finetuning. Without loss of generality, assume $r \ll n$ and let the true transformation matrix $W_t^* \in \mathbb{R}^{r \times n}$ be interpreted as $r$ $n$-dimensional vectors: $\mathbf{w}_t^{*1}, ..., \mathbf{w}_t^{*r} \in \mathbb{R}^n$. Finding LoRA weights is equivalent to finding sets of these $r$ vectors in $\mathbb{R}^n$.

**Definition 3.2** (Set of LoRA weights). We define the weights of a LoRA adapted for task $t_i$ as $B_i A_i$. Both $B_i$ and $A_i$ will have their own individual subspaces. For the purpose of the analysis we will consider a generic task specific weight matrix $W_i \in \mathbb{R}^{m \times n}$ adapted to task $t_i$ such that $n < m$ and its rank $r < n$. The analysis, however, is valid for both $B_i$ and $A_i$. Now we can define a set of LoRAs as stacked (along columns) weight matrices $\hat{W} = \{W_i\}_{i=1}^d$ where each $W_i$ is adapted for a task $t_i \in \mathcal{T}_d$ and a training set $\mathcal{S}_i = \{\{x, y\} \mid x \in X_t, y \in Y_t\}$ where the size of the training is $s_i = |\mathcal{S}_i|$. For theoretical analysis we assume that each training set $X_i \times Y_i$ is distributed according to some unknown Gaussian distribution with mean $\bar{X}_i$ and $\|X_i\|_F \le M$ for some constant $M > 0$. Each weight matrix can have different ranks and the following method and analysis will still hold; however, for brevity we assume all weight matrices stacked in $\hat{W}$ to have the same rank $r$.

**Definition 3.3** (Subspace spanned by LoRAs from a task domain $\mathcal{T}_d$). We define the subspace of weights $\mathcal{Z}_d = \{C\hat{W} \mid C \in \mathbb{R}^{m \times m}\}$ spanned within $\mathbb{R}^{m \times n}$.

Using Singular Value Decomposition (SVD) or Principal Component Analysis (PCA for a zero-centered $\hat{W}$), we can obtain $\hat{W} = \mathcal{U}\Sigma\mathcal{V}^T$. We then represent the top $K$ right singular vectors of $\hat{W}$ (or top $K$ principal components if $\hat{W}$ is zero-centered) as $\mathcal{V}_K^T \in \mathbb{R}^{K \times n} = \{\mathcal{V}_k \in \mathbb{R}^{1 \times n}\}_{k=1}^K$.

**Definition 3.4** (Shared principal subspace of LoRAs finetuned in domain $\mathcal{T}_d$). We define the shared principal subspace of weights for a task domain $\mathcal{T}_d$ as $\mathcal{Z}_d^K = \{\alpha\mathcal{V}_K^T \mid \alpha \in \mathbb{R}^{m \times K}\}$ spanned by top K principal components of the LoRAs within $\mathbb{R}^{m \times n}$.

Next, we introduce idea of defining a new related task $t_{d+1}$

**Definition 3.5** (New related task $t_{d+1}$). . A new linear task $t_{d+1}$ with true solution $W^*_{d+1}$ is said to be related if it is spanned by the basis of $\hat{W}$ i.e. $W^*_{d+1} = C\hat{W}$ and it holds that $\|W^*_{d+1} - \alpha^*_{d+1}\mathcal{V}_K^T\|_F^2 \leq \|W^*_{d+1} - \alpha_{d+1}\mathcal{V}_K^T\|_F^2$ for all rank $K$ linear transformation matrices $\alpha_{d+1}$ and $\|W^*_{d+1} - \alpha^*_{d+1}\mathcal{V}_K^T\|_F^2 \leq \|C\|_2^2 \sum_{i=K+1}^{nd} \sigma_i^2$ where $\sigma_i$'s are singular values of $\hat{W}$. For such a task, we learn coefficients of $K$ principal components $\alpha_{d+1} \in \mathbb{R}^{m \times K}$ resulting in EigenLoRAx weights $W_{\mathcal{E}} = \alpha_{d+1}\mathcal{V}_K^T$.

Definition 3.5 establishes a bound over the related-ness of a new task with those in the known task domain $\mathcal{T}_d$. If the true solution of the new task lies majorly in the principal subspace of $\mathcal{T}_d$ i.e. has major principal components (PCs) within the top $K$ principal components of $\hat{W}$ with some finite bound on the misalignment along the PCs orthogonal to the top $K$ PCs of $\hat{W}$, then we can ideally quantify the relation between a new task and a task domain. Any task that has its true solution within a subspace defined by the PCs orthogonal to the top $K$ PCs of $\hat{W}$ is not as closely related as a task with its solution completely or majorly within the principal subspace. A task that has its solution completely orthogonal to all the PCs of $\hat{W}$ is completely unrelated and is not our main focus here.

Next, we present an algorithm to find the principal subspace and our experiments in Section 4.

## 3.2 ALGORITHM

Assume we have $N$ LoRA adapters, each consisting of a set of $A, B$ matrix pairs for every layer, trained on various tasks within a domain $\mathcal{T}_d$ for a given base pretrained model. Algorithm 1 computes a list of top $K$ principal components—referred to as EigenLoRAx PCs—that define an initial principal subspace for this domain.

To construct this subspace, the algorithm aggregates LoRA matrices across tasks for each layer, separately for $A$ and $B$ matrices (though it can also be applied to the product $BA$). Each LoRA matrix, having rank r, is treated as a list of vectors, and a decomposition is performed on this stacked set of vectors. The most significant components extracted from this process serve as a basis for the principal subspace, providing an efficient representation that can be linearly combined to approximate the original LoRA weight matrices. We showcase our algorithm using representative weight matrices $W_t$, where each $W_t$ represents a single $A$ or $B$ matrix from a single LoRA layer of the neural network. In practice, this procedure is applied to all relevant layers.

Since real-world scenarios often involve low-resource domains with limited availability of LoRA adapters, we extend our subspace by introducing additional pseudo-PCs. Specifically, we sample random vectors of the same dimension as each PC and orthogonalize them with respect to all existing PCs. This process can be iterated to generate more pseudo-PCs, thereby augmenting the principal subspace. As empirically shown in Table 3, this augmentation strategy significantly outperforms naive random selection of PCs for subspace expansion.

**Learning new tasks** Having extracted a set of PCs (including pseudo-PCs, if needed), $\mathcal{V}_K \in \mathbb{R}^{K \times n} = \{\mathcal{V}_k \in \mathbb{R}^{1 \times n}\}_{k=1}^K$, we can approximate a given (LoRA) weight matrix by minimizing $\|W - \alpha\mathcal{V}_K^T\|_F$ where $\alpha$ are linear coefficients Section 3.1. In fact, we can analytically compute of the given LoRA matrices by calculating the linear coefficients which minimizes the above objective. For new tasks however, for which we do not have a LoRA matrix, we freeze the EigenLoRAx PCs and randomly initialize the $\alpha$s. The forward pass in layer is calculated as

$$h = W_0 x + \alpha_B^T \mathcal{V}_B \alpha_A^T \mathcal{V}_A(x). \tag{1}$$

Here, $W_0$ are the pretrained weights of the base model and $\mathcal{V}_B, \mathcal{V}_A$ are EigenLoRAx components (which represent the shared subspace) that are frozen during training. The corresponding lightweight coefficients $\alpha_B$ and $\alpha_A$ are learned. This reduces the number of learnable parameters from $O(2rn)$ to $O(2K)$, by a factor of $\frac{rn}{K}$ (assuming $\alpha$ to be scalar).

Using the definitions 3.1, 3.2, 3.5, 3.4 and 3.3 we state the following theorem;

**Theorem 3.6.** *For a task $t_{d+1}$, we assume a hypothesis $h \in \mathcal{H}_{W_{d+1}}$ expressed as $h(W_{d+1}, X) = W_{d+1}X_{d+1} + W_0X_{d+1} + b$ where $W_{d+1}$ has rank $m$, $b$ is some constant and $W_0$ represents weights of a pretrained foundation model that is frozen during finetuning respectively. We have $h^{\mathcal{E}} \in \mathcal{H}_{W_{\mathcal{E}}}, h^* \in \mathcal{H}_{W^*_{d+1}}$ such that $h^{\mathcal{E}}(W_{\mathcal{E}}, X_{d+1}) = \alpha_{d+1}\mathcal{V}_K^T X_{d+1} + W_0X_{d+1} + b$ where $W_{\mathcal{E}}$ has rank $K$ and $h^*(W^*_{d+1}, X_{d+1}) = C\hat{W}X_{d+1} + W_0X_{d+1} + b$ where $h^*(W^*_{d+1}, X_{d+1}) = Y_{d+1}$ is the true solution for task $t_{d+1}$. For a Lipschitz continuous loss $(\ell^F_{\mathcal{S}_t}(h))$ that is strong convex within the shared principal subspace spanned by principal components $\mathcal{V}_K^T$ with some Lipschitz constant $(L)$, the*

### Algorithm 1 EigenLoRAx PCs Calculation

**Input:** LoRA matrices $\{W_t \in \mathbb{R}^{m \times n}\}_{t=1}^d$, number of PC $(K)$, number of pseudo-PC $(P)$
**Output:** EigenLoRAx PCs $\mathcal{V}_K^T$
$\hat{W} = \begin{bmatrix} W_1 \in \mathbb{R}^{m \times n} & ... & W_d \in \mathbb{R}^{m \times n} \end{bmatrix}$, {Stack LoRA matrices}
Compute the mean of each feature: $\bar{W} = \frac{1}{n}\sum_{i=1}^n W_i$
Subtract the mean: $\hat{W}_c = \hat{W} - \bar{W}$
Perform SVD: $\hat{W}_c = U\Sigma V^T$
Extract the top $K$ principal components
Select the first $K$ columns of $\mathcal{V}$: $\mathcal{V}_K = V[:, 1:K]$
Optionally, augment the subspace with $P$ pseudo-PCs
**for** $p = 1$ to $P$ **do**
    Sample a random vector $v_p \sim \mathcal{N}(0, I_n)$ {Sample from a normal distribution}
    Orthogonalize $v_p$ against all PCs in $\mathcal{V}_K$ using Gram-Schmidt:
    **for** $i = 1$ to $K + p - 1$ **do**
        $v_p = v_p - \frac{v_p^T \mathcal{V}_K[:,i]}{\|\mathcal{V}_K[:,i]\|^2}\mathcal{V}_K[:,i]$
    **end for**
    Normalize $v_p$: $v_p = \frac{v_p}{\|v_p\|}$
    Append $v_p$ to $\mathcal{V}_K$ if $v_p$ is not a null vector
    $K = K + 1$
**end for**
return $\mathcal{V}_K, \mu$

*risk can be written as $R^F_{\mathcal{S}_{d+1}}(h_W) = E_{\mathcal{S}_t}[\ell^F_{\mathcal{S}_t}(h)]$, and using Rademacher complexity bounds we can say with probability at least $1 - 4\delta$ for some $\delta > 0$,*

$$\|W^*_{d+1} - W_{d+1}\|_F^2 \leq C_1 \cdot \left(\frac{\sqrt{m}}{\sqrt{s_t}}\right) + C_2 \tag{2}$$

$$\|\alpha^*_{d+1}\mathcal{V}_K^T - W_{\mathcal{E}}\|_F^2 \leq C_1 \cdot \left(\frac{\sqrt{K}}{\sqrt{s_t}}\right) + \|C\|_2^2 \sum_{i=K+1}^{nd} \sigma_i^2 + C_2 \tag{3}$$

*where $\sigma_i$ are singular values of $\hat{W}$, $C$ is some constant such that $W^*_{d+1} = C\hat{W}$ and $C_1$, $C_2$ are some constants.*

Theorem A.1 provides an upper bound on the Frobenius norm of the difference between $W_{d+1}$ or $W_{\mathcal{E}}$ and the optimal solution $W^*_{d+1}$. 5 provides a tighter upper bound on the norm of the difference when task $t_{d+1}$ majorly lies in the shared principal subspace. The extent to which task $t_{d+1}$ lies in the shared principle subspace is captured by the second term involving the sum of squared truncated singular values $\hat{W}$. Hence, if the task completely or majorly lie in the shared principal subspace, then the first term (sqrt(rank)) will dominate the upper bound. Hence, if rank($W_{d+1} \geq K$), then we can see that the upper bound in eq. 5 will be tighter than in eq. 4 where the task lies majorly in the shared principal subspace. Similarly, when $m \leq K$, the upper bound on the difference norm will be tighter for $W_{d+1}$ than $W_{\mathcal{E}}$. When $W^*_{d+1}$ has a significant alignment or projection along the singular vectors orthogonal to the ones with top $K$ singular values, then the second term in 4 comes into picture and it becomes difficult to directly compare the bounds in 4 and 5. However, if majority of the variance of $W^*_{d+1}$ is along the singular vectors orthogonal to the top $K$ components, it follows that $W_{\mathcal{E}}$ will never be able to achieve convergence while $W_{d+1}$. In contrast, $W_{d+1}$ could perform significantly better, as it is not restricted to learning only along the top $K$ principal components of $\hat{W}$.

**How to choose optimal number of PCs** $K$   The hyperparameter $K$, which determines the number of top PC, can be viewed as a function of task domain complexity—simpler domains require a smaller $K$, while more complex domains benefit from a larger $K$. In practice, we determine $K$ based on empirical observations (Appendix B), evaluating performance across different values. Additionally, we can leverage established techniques from literature, such as explained variance and singular value thresholds Gavish & Donoho (2014). As illustrated in Figure 2, most of the relevant information is often concentrated in a few top EigenLoRAx PCs, providing a practical criterion for selecting $K$.

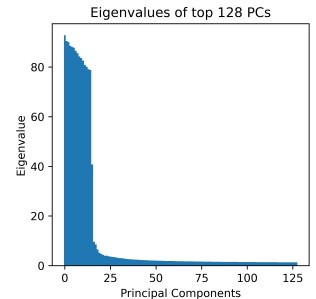

Figure 2: The top 16 components contain the most information from a total of 4000+ components for 500 LoRAs. ($A$ matrices from layer 1 of Mistral-7b model, Lots of LoRAs, see Section 4.3).

**Memory Efficiency and Complexity**   Our method demonstrates significant memory efficiency across experiments. A single set of EigenLoRAx PCs, combined with lightweight task-specific coefficients, can effectively replace both past and future LoRAs within a task domain. This is particularly advantageous when serving a large number of adapters, where frequent loading and unloading in VRAM incurs high latency, or keeping all adapters in memory demands excessive VRAM. For $d$ LoRAs of rank $r$ and $l$ layers, the memory footprint is $O(2drln)$. For EigenLoRAxs, it is $O(2Kl(d+n))$. As $r, K \ll n$, EigenLoRAx becomes more memory efficient in terms of memory required to save the models as $d$ increases. This becomes significantly useful for edge devices and large scale user serving AI systems.

## 4 EXPERIMENTS AND RESULTS

In this section, we demonstrate the efficacy and versatility of EigenLoRAx across diverse tasks, modalities, and model architectures, highlighting its individual advantages. EigenLoRAx requires significantly fewer parameters to match or surpass LoRA's performance (Tables 1, 2) and achieves similar or faster loss convergence (Figure 3), making it a cost-effective alternative to random initialization and other methods (Meng et al., 2024). Additionally, we showcase its memory-efficient inference capabilities with a Stable Diffusion text-to-image generation model (Rombach et al., 2021) (Section 4.4). Notably, EigenLoRAx retains its efficiency even in low-resource scenarios where a large number of LoRAs are unavailable.

**Note on Baselines** Our focus is on recycling adapter knowledge and improving training and memory efficiency while maintaining performance, not solely on maximizing performance. We compare primarily with LoRA, as EigenLoRAx builds its principal subspace using pretrained LoRA adapters. Using better adapters and optimization could further enhance the subspace and performance.

See more experiments (3D Object pose estimation) and detailed ablation experiments in Appendix A.

Table 1: Image classification with Vision Transformer. ZS refers to zero-shot. AUG refers to Augmented for Low-Resource. EigenLoRAx matches or increases performance with drastically fewer number of parameters.

|  | # TRAIN PARAMS | CIFAR 100 | FOOD 101 | FLOWERS 102 |
|---|---|---|---|---|
| FULL TRAINING | 86M | 97.0 | 96.64 | 98.82 |
| BASE MODEL | 15K | 90.07 | 90.8 | 80.71 |
| LORA ($r = 4$) | +147K | 93.79 | **95.73** | 95.03 |
| LORA ($r = 1$) | +36K | 92.45 | 91.07 | 90.14 |
| VERA | +18K | 90.87 | 91.75 | 91.25 |
| ELORAX[AUG] | +1K | 94.4 | 95.01 | 97.5 |
| ELORAX | **+96** | **94.8** | 95.14 | **98.44** |
| ELORAX[ZS] | +0 | 91.4 | 92.48 | 95.7 |

### 4.1 IMAGE CLASSIFICATION

This simpler task involves related datasets where the LoRAs used to construct EigenLoRAx are well-aligned with the downstream tasks, highlighting its finetuning efficiency.

**Setup**   We evaluate EigenLoRAx using a pretrained ViT (ViT) (Dosovitskiy et al., 2021) across 3 datasets. Each dataset is partitioned into 5–6 non-overlapping sub-datasets, mimicking continual learning (Kaushik et al., 2021) and federated learning (Shenaj et al., 2023) setups. As the sub-datasets are derived from the same source, their tasks are more domain-aligned. For EigenLoRAx, we compute

principal components (PCs) using all but one LoRA trained on individual sub-datasets (leave-one-out approach, Algorithm 1). The coefficient matrix $\alpha$ for the excluded task is then learned as described in Section 3.2. All methods are finetuned for 10 epochs, with additional details in Appendix A.2.

**Parameter Efficiency** Table 1 summarizes our experimental results. All models require training the last linear layer (approx. 15K parameters) due to the pre-trained ViT having a different number of categories. For the Base Model, no additional parameters are trained. EigenLoRAx adapts to new sub-datasets using only two principal components (96 additional parameters), enabling it to match or outperform LoRA and VeRA, which use significantly more parameters. We also tested a zero-shot EigenLoRAx (weight initialized randomly within the principal subspace), training only the last layer. This model outperforms the base model with no additional parameters, demonstrating the effectiveness of principal subspace extraction. We also test a low resource scenario (ELoRAx$^{\text{AUG}}$), where only 2 LoRAs are available for extracting the PCs, which are then augmented using random, orthogonal PCs as described in Algorithm 1.

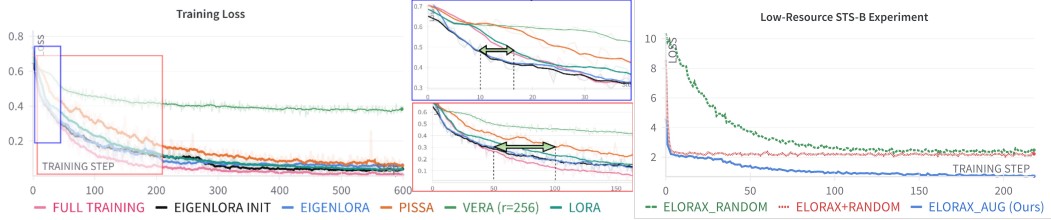

Figure 3: **Fast Convergence and Better Initialization** (left) EigenLoRAx demonstrates faster convergence compared to LoRA and VeRA. EigenLoRAx achieves a speedup of up to $1.5\times$ against LoRA and up to $2\times$ compared to PISSA. This experiment was carried out on the CoLA task of the GLUE benchmark.

## 4.2 GLUE BENCHMARK

Table 2: GLUE benchmark results. We report Matthew's correlation for CoLA, Pearson correlation for STS-B, and accuracy for the remaining tasks. In all cases, higher values indicate better performance.

| METHOD | # TRAINABLE PARAMETERS | MRPC | SST-2 | CoLA | QNLI | RTE | STS-B | AVG. |
|---|---|---|---|---|---|---|---|---|
| FULL TRAINING | 125M | 88.97 | 91.28 | 59.81 | 92.29 | 79.78 | 90.89 | 83.84 |
| PISSA [36] | 1.2M | 86.52 | 94.15 | 61.32 | 92.15 | 71.84 | 90.25 | 82.70 |
| EIGENLORAx$^{\text{INIT}}$ | 1.2M | 89.71 | 93.35 | 61.58 | 92.2 | 74.73 | 89.56 | 83.52 |
| LoRA ($r = 32$) | 1.2M | 86.76 | **94.72** | 59.56 | 92.53 | 77.61 | **90.81** | **83.67** |
| VeRA ($r = 256$) | 25K | 75.98 | 93.23 | 54.14 | 89.21 | 66.78 | 87.03 | 77.72 |
| EIGENLORAx | **12K** | 87 | 94.15 | **59.81** | 92.73 | 77.62 | 90.58 | 83.65 |

Next, we evaluate EigenLoRAx on the General Language Understanding Evaluation (GLUE) benchmark (Wang et al., 2019) datasets using the RoBERTa$_{base}$ model (Liu et al., 2019). We use 6 different tasks: MRPC, SST-2, CoLA, QNLI, RTE and STS-B. Following the setup of VeRA, we omit time-intensive MNLI and QQP tasks, thus avoiding the use of MNLI initialization for MRPC, RTE, and STS-B tasks. In this setting, LoRAs are trained not on sub-datasets but on these different datasets representing a *heterogeneous* domain setting, where the domain difference may be larger relative to the more domain-aligned setting in Section 4.1. We follow the previous leave-one-out setup, where EigenLoRAx PCs are calculated using LoRAs of all but one task, and $\alpha$ is learnt for the left-out task. Refer to Appendix A.3 for all hyperparameters and implementation details.

**Faster Convergence** Our results in Table 2 show that EigenLoRAx ($K = 32$) matches LoRA's performance with **100$\times$ fewer trainable parameters** and outperforms VeRA. EigenLoRAx extracts a useful principal subspace across diverse domains, enabling robust adaptation to new tasks. We also evaluate EigenLoRAx$^{(\text{init})}$ weight initialization speed-up. Unlike PiSSA (Meng et al., 2024), which initializes LoRA matrices with principal directions of pretrained weights, we randomly initialize weights within our extracted subspace. As shown in Figure 3, EigenLoRAx converges faster than PiSSA and VeRA, and slightly faster than LoRA, highlighting the effectiveness of the principal subspace. VeRA's poorer performance may stem from suboptimal random initialization that fails to align with task-critical components. ELoRAx is also more efficient in terms of floating point operations for both forward and backward pass, as shown in Table 14.

**Low-Resource Scenario** To demonstrate the effectiveness of our subspace augmentation strategy Algorithm 1, we conduct an experiment where EigenLoRAx is initialized with only 1–2 LoRAs. The results are presented in Table 3. We compare our method against augmenting EigenLoRAx with random components (EigenLoRAx+random) and using entirely random components (ELoRAx$^{\text{random}}$). As shown, our augmentation approach significantly outperforms random principal component selection. Interestingly, for MRPC, the base model's performance is retained. This suggests that the learned LoRA weights may not have influenced the base model, likely because they did not capture relevant information.

While we do not provide theoretical guarantees for our principal component augmentation strategy—where randomly sampled vectors are iteratively orthogonalized to the existing EigenLoRAx principal vectors—we hypothesize that this targeted guidance helps prevent redundancy within the subspace. Consequently, it increases the likelihood of capturing the necessary task-relevant components.

Table 3: Low-Resource GLUE Subset Results

|  | # PARAM | MRPC | STS-B |
| --- | --- | --- | --- |
| ELoRAx$^{\text{RANDOM}}$ | 24K | 68.38 | -0.73 |
| ELoRAx+RAND | 24K | 68.38 | 0.11 |
| ELoRAx$^{\text{AUG}}$ | 24K | 83.09 | 85.28 |

Table 4: Lots of LoRAs. We report the Rouge-L scores for each of the 5 tasks from the training set and 5 from the testing set. EigenLoRAx achieves on average **88%** of LoRA's performance while requiring anywhere from $12\times$ to $95\times$ less parameters in a zero-shot setting.

| METHOD | # TRAINABLE PARAMETERS | 076 | 627 | 664 | 819 | 1631 | 039 | 290 | 391 | 442 | 1598 | AVG. |
| --- | --- | --- | --- | --- | --- | --- | --- | --- | --- | --- | --- | --- |
| LoRA ($r = 16$) | 9.4M | 69.05 | 23.96 | 25 | 75 | 99.04 | 58.77 | 93.79 | 93.45 | 67.84 | 51.58 | 65.75 |
| EIGENLORAX$^{\text{ZS}}$ | **98-786K** | 60.78 | 18.91 | 33.33 | 65.07 | 94.74 | 49.96 | 84.54 | 88.56 | 49.78 | 39.81 | 58.25 |
| PERFORMANCE RATIO |  | 0.88 | 0.79 | 1.33 | 0.87 | 0.96 | 0.79 | 0.90 | 0.95 | 0.73 | 0.77 | 0.88 |

## 4.3 LOTS OF LoRAs

Finally, we also tested our method in settings where a large number of adapters may be trained on significantly diverse domains. Lots of LoRAs (Brüel-Gabrielsson et al., 2024) is a collection of over 500 adapters of the Mistral-7B-Instruct-v0.2 model (Jiang et al., 2023), trained on a variety of natural instruction tasks (Wang et al., 2022). It represents the realistic setting where we directly use publicly available trained adapters, which may present significant diversity in terms of quality and task domain. As all adapters are accompanied with their respective training datasets, Lots of LoRAs is particularly useful in evaluating EigenLoRAx. The task presents significant diversity and a higher $K$ is necessary to represent this open domain.

**Setup** We split adapters randomly into two sets (490,5). EigenLoRAx PCs were calculated using the larger "training" set and evaluations were done on the smaller "test" set. We evaluated EigenLoRAx in a zero-shot setting (calculated using the already available adapter weights, no finetuning) . The results are shown in Table 4 where we evaluate EigenLoRAx on the 5 tasks from the test set and also on 5 tasks from the training set to check for catastrophic forgetting or concept drift from scaling. The first 5 tasks are randomly sampled from the training set. **EigenLoRAx nearly matches LoRA**

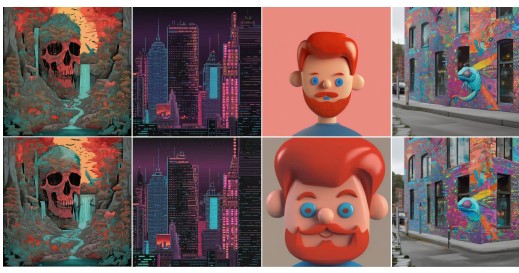 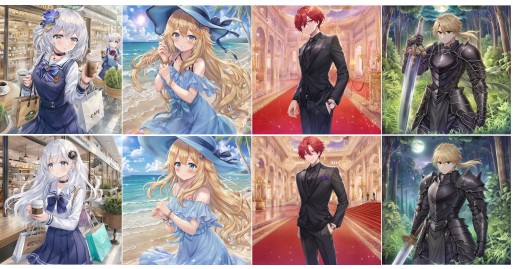

Figure 4: LoRAs (top) vs. EigenLoRAx (bottom) in Text-to-Image generation. (Left) A single EigenLoRAx analytically reconstructs multiple LoRAs, significantly reducing memory ($18\times$ reduction) and compute costs. (Right) It efficiently learns new tasks with up to $100\times$ fewer parameters than LoRA, maintaining similar visual quality. See Appendix A.4 for more examples.

**with** $12 - 95\times$ **fewer parameters**. EigenLoRAx recovers upto 88% of LoRA's performance even

in a zero-shot setting at such a large scale. The performance of EigenLoRAx can be improved by fine-tuning the EigenLoRAx adapters. In this setting we use randomized SVD in order to speed up the calculation of the PCs. We believe this leads to some degradation in performance as there randomized methods are approximations of the actual calculations. Performance can be further improved if better implementations of SVD which do not sacrifice accuracy for speed are used in calculating the Principal Components.

## 4.4 TEXT-TO-IMAGE IMAGE GENERATIVE MODELS

We showcase EigenLoRAx's versatility on complex multimodal tasks like text-to-image generation, where LoRAs are extensively used to adapt models like Stable Diffusion to various styles and datasets. Despite thousands of LoRA adapters being available, most remain underutilized, occupying significant memory alongside their data. As adapter usage grows, a critical challenge is efficiently hosting multiple adapters for diverse tasks, especially on edge devices. Switching adapters during inference, often from CPU memory or disk, introduces latency that hinders real-time applications. EigenLoRAx tackles this by extracting a shared task-invariant subspace, significantly reducing in-memory parameters and enabling memory-efficient inference without compromising flexibility or performance. EigenLoRAx can effectively replace pretrained adapters, drastically reducing storage requirements. To demonstrate this, we extracted $K = 14$ principal components from $N = 20$ Stable Diffusion-XL (Podell et al., 2023) LoRA adapters (rank $r = 32$) from the HuggingFace diffusers library (von Platen et al., 2022). Using $\alpha \in \mathbb{R}^{r \times K}$, we analytically reconstructed the original LoRA weights within the extracted principal subspace. For image generation, we used 30 denoising steps with a fixed seed of 0. Results and comparisons are shown in Figure 4. This approach reduces storage requirements for all adapters from 4.6GB to just 261MB, achieving an **18× reduction in low-rank parameters stored in memory**. By enabling a large number of adapters to reside in VRAM simultaneously, EigenLoRAx eliminates I/O bottlenecks, significantly improving memory efficiency for real-time applications.

**Failure Cases and Limitations** Despite its advantages, EigenLoRAx has limitations. Appendix Figure 9 shows a failure case where the method fails to capture a key property of the desired image. While tasks may share a principal subspace, missing critical orthogonal components can degrade performance, especially if they were absent in the pretrained LoRAs used for extraction or if the chosen top $K$ components were suboptimal. In the latter case, empirical analysis of hyperparameters (Appendix B) can guide optimal $K$ selection. Additionally, our subspace augmentation method (Table 3) helps by iteratively sampling and orthogonalizing more components to recover missing subspace elements. A simple extension can further mitigate this issue by allowing a small number of rank-1 weights to be trainable outside the subspace. Another key limitation (Section 4.3) is the computational cost and instability of processing a large number of initial LoRAs. A continual learning approach building on our method could address this. Finally, our experiments did not explore layer-wise or weight matrix-level optimizations; we tested different $K$ values but kept them fixed across layers and for both A and B matrices. Additional failure cases are discussed in Appendix B.2.

## 5 CONCLUSION

We introduce EigenLoRAx, a significantly efficient model finetuning and inference method that recycles publicly available pretrained adapters by finding a shared principal subspace. This allows finetuning on new data by simply learning the lightweight coefficients of the shared subspace, and also requires less number of parameters to be saved for new tasks. Our comprehensive and diverse experiments show that EigenLoRAx is applicable to a large range of problems and model architectures. We believe that EigenLoRAx has the potential to mitigate the perpetually widening compute resource gap (Ahmed & Wahed, 2020; Besiroglu et al., 2024) and reduce the environmental cost of training and using machine learning models (Wu et al., 2021; Ligozat et al., 2021). It also holds promise for training personalized models (Tan et al., 2024) on low-resource devices, in privacy-critical use-cases. We have a large number of experiments (6+) on a diverse set of complex models, tasks and modalities. We have shown that EigenLoRAx excels in faster and efficient learning, memory savings and zero shot performance which differentiates it from conventional PEFT models.

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

# A  APPENDIX

## A.1  THEORETICAL ANALYSIS

Using the definitions 3.1, 3.2, 3.5, 3.4 and 3.3 we state the following theorem;

**Theorem A.1.** *For a task $t_{d+1}$, we assume a hypothesis $h \in \mathcal{H}_{W_{d+1}}$ expressed as $h(W_{d+1}, X) = W_{d+1}X_{d+1} + W_0X_{d+1} + b$ where $W_{d+1}$ has rank $m$, $b$ is some constant and $W_0$ represents weights of a pretrained foundation model that is frozen during finetuning respectively. We have $h^{\mathcal{E}} \in \mathcal{H}_{W_{\mathcal{E}}}, h^* \in \mathcal{H}_{W^*_{d+1}}$ such that $h^{\mathcal{E}}(W_{\mathcal{E}}, X_{d+1}) = \alpha_{d+1}\mathcal{V}_K^T X_{d+1} + W_0 X_{d+1} + b$ where $W_{\mathcal{E}}$ has rank $K$ and $h^*(W^*_{d+1}, X_{d+1}) = C\hat{W}X_{d+1} + W_0X_{d+1} + b$ where $h^*(W^*_{d+1}, X_{d+1}) = Y_{d+1}$ is the true solution for task $t_{d+1}$. For a Lipschitz continuous loss $(\ell_{\mathcal{S}_t}^F(h))$ that is strong convex within the shared principal subspace spanned by principal components $\mathcal{V}_K^T$ with some Lipschitz constant $(L)$, the risk can be written as $R_{\mathcal{S}_{d+1}}^F(h_W) = E_{\mathcal{S}_t}[\ell_{\mathcal{S}_t}^F(h)]$, and using Rademacher complexity bounds we can say with probability at least $1 - 4\delta$ for some $\delta > 0$,*

$$\|W^*_{d+1} - W_{d+1}\|_F^2 \leq C_1 \cdot \left(\frac{\sqrt{m}}{\sqrt{s_t}}\right) + C_2 \tag{4}$$

$$\|\alpha^*_{d+1}\mathcal{V}_K^T - W_{\mathcal{E}}\|_F^2 \leq C_1 \cdot \left(\frac{\sqrt{K}}{\sqrt{s_t}}\right) + \|C\|_2^2 \sum_{i=K+1}^{nd} \sigma_i^2 + C_2 \tag{5}$$

*where $\sigma_i$ are singular values of $\hat{W}$, $C$ is some constant such that $W^*_{d+1} = C\hat{W}$ and $C_1$, $C_2$ are some constants.*

*Proof.* The derivation is straightforward, we can write the difference in risks for $h^{\mathcal{E}}$ and $h^*$ as

$$R_{\mathcal{S}_{d+1}}^F(h^{\mathcal{E}}) - R_{\mathcal{S}_{d+1}}^F(h^*) = \mathbb{E}_{\mathcal{S}_t}\left[\ell_{\mathcal{S}_t}^F(h^{\mathcal{E}}) - \ell_{\mathcal{S}_t}^F(h^*)\right]$$

By definition of strong convex loss function for some constant $\mu \geq 0$,

$$\mathbb{E}_{\mathcal{S}_t}\left[\ell_{\mathcal{S}_t}^F(h^{\mathcal{E}}) - \ell_{\mathcal{S}_t}^F(h^*)\right] \geq \frac{\mu}{2}\|W_{\mathcal{E}} - W^*_{d+1}\|_F^2$$

We also know from generalization error bounds using Rademacher Complexity from Bartlett & Mendelson (2003) that with probability at least $1 - 2\delta$,

$$|R_{\mathcal{S}_{d+1}}^F(h^{\mathcal{E}}) - \hat{R}_{\mathcal{S}_{d+1}}^F(h^{\mathcal{E}})| \leq \frac{\mathcal{R}_{s_{d+1}}(\mathcal{H}_{W_{\mathcal{E}}})}{2} + \sqrt{\frac{\ln(1/\delta)}{2s_t}}$$

We can rewrite risk as

$$\begin{aligned} R_{\mathcal{S}_{d+1}}^F(h^*) - R_{\mathcal{S}_{d+1}}^F(h^{\mathcal{E}}) =\; & R_{\mathcal{S}_{d+1}}^F(h^*) - \hat{R}_{\mathcal{S}_{d+1}}^F(h^*) \\ & - R_{\mathcal{S}_{d+1}}^F(h^{\mathcal{E}}) + \hat{R}_{\mathcal{S}_{d+1}}^F(h^{\mathcal{E}}) \\ & + \hat{R}_{\mathcal{S}_{d+1}}^F(h^*) - \hat{R}_{\mathcal{S}_{d+1}}^F(h^{\mathcal{E}}) \end{aligned}$$

Since we know by definition of $h^*$ that $\hat{R}_{\mathcal{S}_{d+1}}^F(h^*) \leq \hat{R}_{\mathcal{S}_{d+1}}^F(h^{\mathcal{E}})$, we can say

$$\begin{aligned} R_{\mathcal{S}_{d+1}}^F(h^*) - R_{\mathcal{S}_{d+1}}^F(h^{\mathcal{E}}) \leq\; & R_{\mathcal{S}_{d+1}}^F(h^*) - \hat{R}_{\mathcal{S}_{d+1}}^F(h^*) \\ & - R_{\mathcal{S}_{d+1}}^F(h^{\mathcal{E}}) + \hat{R}_{\mathcal{S}_{d+1}}^F(h^{\mathcal{E}}) \end{aligned}$$

Then we take a union bound to conclude that with probability at least $1 - 4\delta$,

$$\begin{aligned} R_{\mathcal{S}_{d+1}}^F(h^*) - R_{\mathcal{S}_{d+1}}^F(h^{\mathcal{E}}) \leq\; & \frac{\mathcal{R}_{s_{d+1}}(\mathcal{H}_{W_{\mathcal{E}}})}{2} + \sqrt{\frac{2\ln(1/\delta)}{s_{d+1}}} \\ & + \frac{\mathcal{R}_{s_{d+1}}(\mathcal{H}_{W^*_{d+1}})}{2} \end{aligned}$$

Hence, we can also say that with probability at least $1 - 4\delta$,

$$\frac{\mu}{2}\|W^*_{d+1} - W_{\mathcal{E}}\|_F^2 \leq \frac{\mathcal{R}_{s_{d+1}}(\mathcal{H}_{W_{\mathcal{E}}})}{2} + \sqrt{\frac{2\ln(1/\delta)}{s_t}} + \frac{\mathcal{R}_{s_{d+1}}(\mathcal{H}_{W^*_{d+1}})}{2} \qquad (6)$$

The Rademacher complexity of a low-rank weight matrix class $\mathcal{H}_{W_{\mathcal{E}}}$ with rank $K$ can be directly bounded using results from Bartlett & Mendelson (2003) as

$$\mathcal{R}_{s_{d+1}}(\mathcal{H}_{W_{\mathcal{E}}}) = \mathcal{O}\left(\frac{\sqrt{K}\|W_{\mathcal{E}}\|_F}{\sqrt{s_t}}\right)$$

We can separate the constants including $\mathcal{R}_{s_{d+1}}(\mathcal{H}_{W^*_{d+1}})$ from 6 and assume that, for a normalised $\|W_{\mathcal{E}}\|$, it is usually bounded, then we can write:

$$\|W^*_{d+1} - W_{\mathcal{E}}\|_F^2 \leq C_1 \cdot \left(\frac{\sqrt{K}}{\sqrt{s_t}}\right) + C_2 \qquad (7)$$

Similarly, we can also say for $W_{d+1}$ that

$$\|W^*_{d+1} - W_{d+1}\|_F^2 \leq C_1 \cdot \left(\frac{\sqrt{m}}{\sqrt{s_t}}\right) + C_2 \qquad (8)$$

This proves 4. Now to further prove 5, we use properties of Frobenius norm,

$$\|W_{\mathcal{E}} - \alpha^*_{d+1}\mathcal{V}_K^T\|_F^2 - \|W^*_{d+1} - \alpha^*_{d+1}\mathcal{V}_K^T\|_F^2$$
$$\leq \|W_{\mathcal{E}} - W^*_{d+1}\|_F^2$$

Then following from the definition of $W^*_{d+1}$, we can say that,

$$\|W_{\mathcal{E}} - \alpha^*_{d+1}\mathcal{V}_K^T\|_F^2 - \|C\|_2^2 \sum_{i=K+1}^{nd} \sigma_i^2 \leq \|W_{\mathcal{E}} - W^*_{d+1}\|_F^2$$

Finally, using the Rademacher complexity bound we provided earlier, we can say that with probability at least $1 - 4\delta$

$$\|\alpha^*_{d+1}\mathcal{V}_K^T - W_{\mathcal{E}}\|_F^2 \leq \|W^*_{d+1} - W_{\mathcal{E}}\|_F^2$$
$$\leq C_1 \cdot \left(\frac{\sqrt{K}}{\sqrt{s_t}}\right) + \|C\|_2^2 \sum_{i=K+1}^{nd} \sigma_i^2 + C_2$$

We can just rewrite $W_{\mathcal{E}} = \alpha^*_{d+1}\mathcal{V}_K^T$ and get the same bound as above for $\|\alpha^*_{d+1} - \alpha_{d+1}\|_F^2$. We can similarly obtain the upper bound for 5

This concludes the proof. $\qquad\qquad\qquad\qquad\qquad\qquad\qquad\qquad\qquad\qquad\qquad\quad\square$

Theorem A.1 provides an upper bound on the Frobenius norm of the difference between $W_{d+1}$ or $W_{\mathcal{E}}$ and the optimal solution $W^*_{d+1}$. 5 provides a tighter upper bound on the norm of the difference when task $t_{d+1}$ majorly lies in the shared principal subspace. The extent to which task $t_{d+1}$ lies in the shared principle subspace is captured by the second term involving the sum of squared truncated singular values $\hat{W}$. Hence, if the task completely or majorly lie in the shared principal subspace, then the first term (sqrt(rank)) will dominate the upper bound. Hence, if rank$(W_{d+1} \geq K)$, then we can see that the upper bound in eq. 5 will be tighter than in eq. 4 where the task lies majorly in the shared principal subspace. Similarly, when $m \leq K$, the upper bound on the difference norm will be tighter for $W_{d+1}$ than $W_{\mathcal{E}}$. When $W^*_{d+1}$ has a significant alignment or projection along the singular vectors orthogonal to the ones with top $K$ singular values, then the second term in 4 comes into picture and it becomes difficult to directly compare the bounds in 4 and 5. However, if majority of the variance of $W^*_{d+1}$ is along the singular vectors orthogonal to the top $K$ components, it follows that $W_{\mathcal{E}}$ will never be able to achieve convergence while $W_{d+1}$. In contrast, $W_{d+1}$ could perform significantly better, as it is not restricted to learning only along the top $K$ principal components of $\hat{W}$. While the

assumption that $W^*_{d+1}$ is spanned by the principal components of the shared principal subspace might appear to be very strong, we empirically observe in table 4 that such an assumption is not impractically far from reality. Particularly, we observe in table 2 that for GLUE benchmark, LoRA adapters trained on 5 diverse tasks shared a principal subspace. We see that EigenLoRAx was able to leverage the principal components of this shared subspace with just 12K training parameters learned for a new 6th task and achieve competitive performance compared to fine-tuning full rank weights with 125M parameters or individual LoRA adaptors with 1.2M parameters, even outperforming them in certain tasks. Similarly, table 4 demonstrates zero-shot performance using only top $K$ principal components of the shared subspace obtained through 500 LoRA adaptors trained on diverse tasks. This further suggests that increasing the number of LoRA adapters enables a richer set of top principal components, effectively spanning the shared subspace and providing broader coverage for new tasks.

## A.2 EXPERIMENTS

For VeRA, LoRA and PiSSA, we experimented with a range of learning rates, from higher to lower, along with three different scheduling approaches: ReduceLRonPlateau, Linear, and Cosine. The hyperparameters that yielded the best average performance were selected for further experimentation. The observed discrepancies with EigenLoRAx hyperparameters are attributable to these methodological choices. Comprehensive hyperparameter tuning for EigenLoRAx was not pursued extensively, as the initially selected hyperparameters, notably a high learning rate paired with ReduceLRonPlateau or Linear, demonstrated satisfactory performance, thereby conserving computational resources.

### A.2.1 IMAGE CLASSIFICATION

**Trainable parameters for EigenLoRAx**   The base model is vit-base-patch16-224. The following are the trainable parameters in ViT (Dosovitskiy et al., 2021) that are trained for EigenLoRAx. We ignore the last linear layer for simplicity since it is trained for all models and baselines and is constant. The loading parameter has the shape of [number of EigenLoRAx PC, 1] (we only have 2 in each EigenLoRAx PC for this experiment). Therefore, the total number of trainable parameters (for the number of components= 2) is 12 (layers)  × 4 (set of parameters per layers)  × 2 (number of trainable parameter per coefficient)  = 96 trainable parameters.

**Hyperparameters**   LoRA (Hu et al., 2021) and VeRA (Kopiczko et al., 2023) implementations are taken from the HuggingFace PEFT (Mangrulkar et al., 2022) library with hyperparameters of the default method. For Food101 (Bossard et al., 2014) experiment, we randomly remove 1 class for ease of compute. Experimental hyperparameters are reported in Table 5 and Table 6.

Table 5: Hyperparameters for LoRA (Hu et al., 2021) and VeRA (Kopiczko et al., 2023) for the Image Classification Experiment

|  | CIFAR100 | Flowers102 | Food101 |
|---|---|---|---|
| Learning Rate | 1e−4 | 1e−4 | 1e−4 |
| Weight Decay | 0.1 | 0.1 | 0.1 |
| Warmup ratio | 0.06 | 0.06 | 0.06 |
| Epochs | 10 | 10 | 10 |
| Number of Subsets | 5 | 6 | 5 |
| Categories/Subset | 20 | 17 | 20 |
| Seed | 42 | 42 | 42 |
| Batch Size | 128 | 64 | 128 |

**Experimental Results**   The experiments were conducted 5 times utilizing randomly generated dataset splits. The mean accuracy values are reported in Table 1. Empirical analysis indicates that without control and annealing of learning rates, the loss for both LoRA and VeRA may diverge or plateau, particularly with high learning rates. Even with the lower learning rate, Full training or LoRA can overfit to the training data without proper regularization. In contrast, no such instability was observed during EigenLoRAx training, where a relatively higher learning rate proved advantageous for rapid convergence.

Table 6: Hyperparameters for EigenLoRAx for the Image Classification Experiment

|  | CIFAR100 | Flowers102 | Food101 |
|---|---|---|---|
| Learning Rate | 1e−2 | 1e−2 | 1e−2 |
| Weight Decay | 0.1 | 0.1 | 0.1 |
| Warmup ratio | 0.06 | 0.06 | 0.06 |
| Epochs | 10 | 10 | 10 |
| Number of Subsets | 5 | 6 | 5 |
| Categories/Subset | 20 | 17 | 20 |
| Seed | 42 | 42 | 42 |
| Batch Size | 128 | 64 | 128 |

Table 7: Image Classification Accuracy results on CIFAR100 (Krizhevsky et al., 2009)

| Model | Trainable Params | subset1 | subset2 | subset3 | subset4 | subset5 | Avg. |
|---|---|---|---|---|---|---|---|
| **FT** | 86389248 | 98.8 | 97.95 | 95.55 | 96.05 | 96.3 | 96.93 |
| **LoRA** ($r = 1$) | 36864 | 97.6 | 93.95 | 93.75 | 91.75 | 85.2 | 92.45 |
| **LoRA** ($r = 4$) | 147456 | 98.15 | 95.2 | 93.5 | 92.85 | 89.25 | 93.79 |
| **VeRA** ($r = 2$) | 18480 | 93.65 | 89.7 | 89.5 | 89.95 | 91.55 | 90.87 |
| **EigenLoRAx** ($K = 2$) | 96 | 97.25 | 95.05 | 94.55 | 93 | 94.15 | 94.8 |

Table 8: Image Classification Accuracy results on Food101 (Bossard et al., 2014)

| Model | Trainable Params | subset1 | subset2 | subset3 | subset4 | subset5 | Avg. |
|---|---|---|---|---|---|---|---|
| **FT** | 86389248 | 98.64 | 97 | 97.36 | 94.28 | 95.92 | 96.64 |
| **LoRA** ($r = 1$) | 36864 | 93.36 | 88.44 | 94.28 | 89.4 | 89.9 | 91.076 |
| **LoRA** ($r = 4$) | 147456 | 98.2 | 96.96 | 96.08 | 92.88 | 94.52 | 95.728 |
| **VeRA** ($r = 2$) | 18480 | 91.22 | 88.42 | 94.42 | 91.88 | 92.82 | 91.752 |
| **EigenLoRAx** ($K = 2$) | 96 | 97.24 | 95.96 | 96 | 91.88 | 94.6 | 95.136 |

Table 9: Image Classification Accuracy results on Flowers102 (Nilsback & Zisserman, 2008)

| Model | subset1 | subset2 | subset3 | subset4 | subset5 | subset6 | Avg. |
|---|---|---|---|---|---|---|---|
| **FT** | 99.7 | 99.3 | 98.01 | 98.22 | 99.7 | 98.01 | 98.82 |
| **LoRA** ($r = 1$) | 85.9 | 88.47 | 92.69 | 91.02 | 91.7 | 91.01 | 90.13 |
| **LoRA** ($r = 4$) | 96.23 | 92.76 | 97.22 | 95.01 | 98.24 | 90.73 | 95.03 |
| **VeRA** ($r = 2$) | 99.2 | 95.4 | 97.7 | 94.7 | 90.9 | 95 | 95.48 |
| **EigenLoRAx** ($K = 2$) | 99.686 | 97.905 | 97.689 | 98.291 | 99.344 | 97.718 | 98.43 |

## A.3 NATURAL LANGUAGE PROCESSING - GLUE BENCHMARK

**Hyperparameters** LoRA (Hu et al., 2021), VeRA (Kopiczko et al., 2023) and PISSA (Meng et al., 2024) implementations are taken from the HuggingFace PEFT (Mangrulkar et al., 2022) library. Refer to Table 10 and Table 11 for hyperparameter details. For LoRA (Hu et al., 2021), we use the ranks $\in \{8, 16\}$. For VeRA (Kopiczko et al., 2023), we use rank$= 256$, and for EigenLoRAx, we use $K \in \{16, 32\}$ and $r = 8$. Here, $r$ refers to the dimensionality of the trainable coefficients and not the rank. For both PISSA (Meng et al., 2024) and LoRA, all the parameters of the low rank matrix are trainable. For the EigenLoRAx initialization experiment, we train both the components and coefficients for a fair comparison with PISSA. In practice, however, we do not need to do so - we can tune only the sparse coefficients and after the loss converges, finetune the components for a few training steps.

Table 10: Hyperparameters for LoRA (Hu et al., 2021), VeRA (Kopiczko et al., 2023) and PiSSA (Meng et al., 2024) for the GLUE benchmark. (Wang et al., 2019)

|  | CoLA | MRPC | QNLI | RTE | SST-2 | STS-B |
|---|---|---|---|---|---|---|
| Learning Rate | 4e−4 | 4e−4 | 4e−4 | 5e−4 | 5e−4 | 4e−4 |
| Weight Decay | 0.1 | 0.1 | 0.1 | 0.1 | 0.1 | 0.1 |
| Warmup ratio | 0.06 | 0.06 | 0.06 | 0.06 | 0.06 | 0.06 |
| Epochs | 80 | 30 | 25 | 80 | 60 | 40 |
| Scheduler | Linear | Linear | Linear | Linear | Linear | Linear |
| Seed | 0 | 0 | 0 | 0 | 0 | 0 |
| Batch Size | 64 | 64 | 64 | 64 | 64 | 64 |

Table 11: Hyperparameters for EigenLoRAx for the GLUE benchmark. (Wang et al., 2019). (RLrP - ReduceLRonPlateau)

|  | CoLA | MRPC | QNLI | RTE | SST-2 | STS-B |
|---|---|---|---|---|---|---|
| Learning Rate | 4e−3 | 4e−3 | 4e−3 | 5e−3 | 5e−3 | 4e−3 |
| Weight Decay | 0.1 | 0.1 | 0.1 | 0.1 | 0.1 | 0.1 |
| Warmup ratio | 0.06 | 0.06 | 0.06 | 0.06 | 0.06 | 0.06 |
| Epochs | 80 | 30 | 25 | 80 | 60 | 40 |
| Scheduler | RLrP | RLrP | RLrP | RLrP | RLrP | RLrP |
| Seed | 0 | 0 | 0 | 0 | 0 | 0 |
| Batch Size | 64 | 64 | 64 | 64 | 64 | 64 |

## A.4 TEXT-TO-IMAGE GENERATION (STABLE DIFFUSION MODELS)

Figure 5 and Figure 6 show more examples of a text-to-image stable diffusion model finetuned using EigenLoRAx. Note that not only there is no publicly available code for VeRA that allows its usage in complex text-to-image generation tasks, but our VeRA implementation also did not work well in this task.

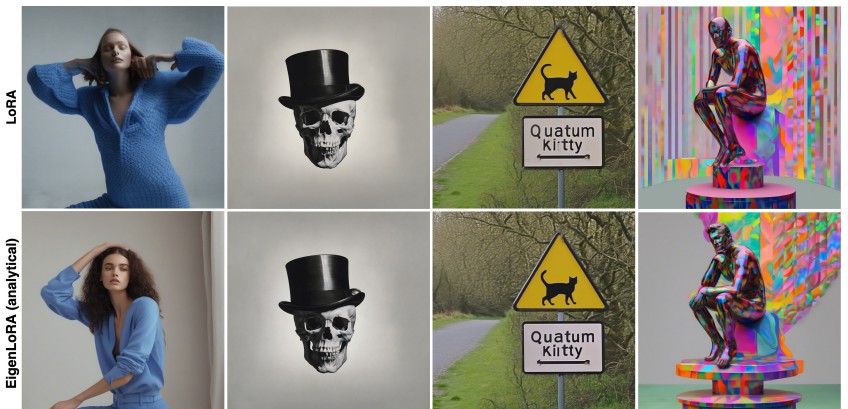

Figure 5: (Part 1) A single EigenLoRAx (identical components, varying loadings) was employed to produce these images utilizing the Stable Diffusion-XL Podell et al. (2023) model. A comparison between our results and those obtained from multiple LoRAs does not show a noticeable degradation in visual quality.

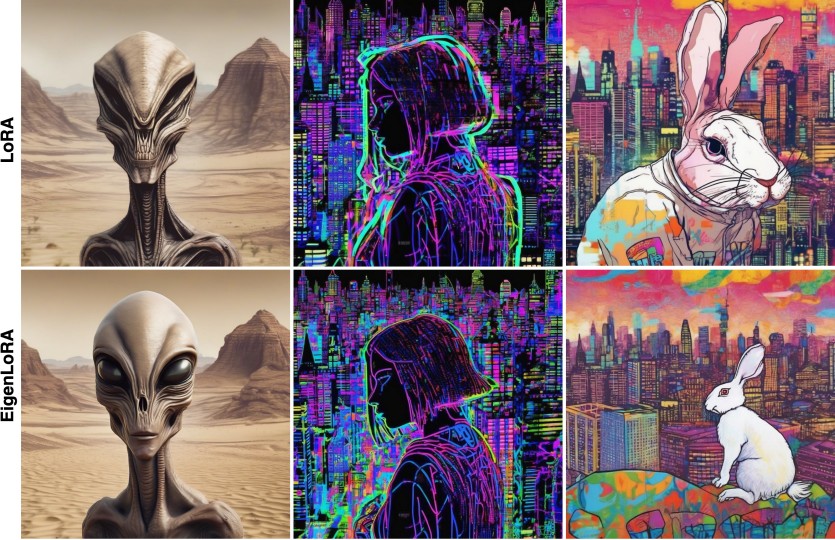

Figure 6: (Part 2) A single EigenLoRAx (identical components, varying loadings) was employed to produce these images utilizing the Stable Diffusion-XL Podell et al. (2023) model. A comparison between our results and those obtained from multiple LoRAs demonstrates no noticeable degradation in visual quality.

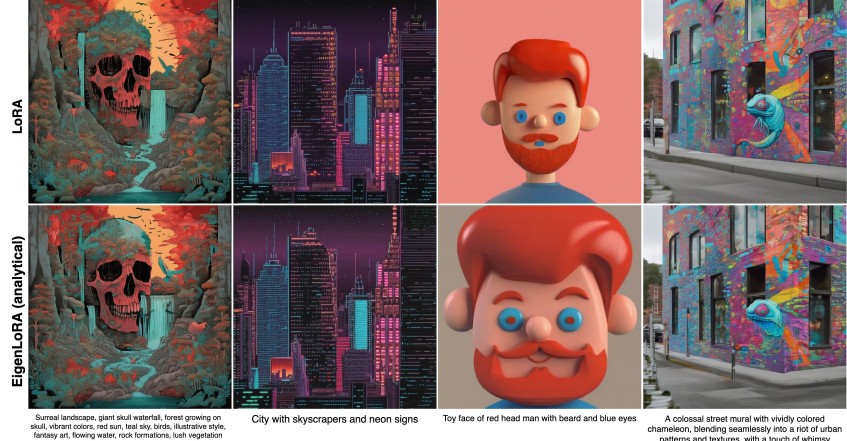

Figure 7: Analytical reconstruction of LoRAs using EigenLoRAx which shows no degradation in relative visual quality. See Appendix A.4 for more examples.

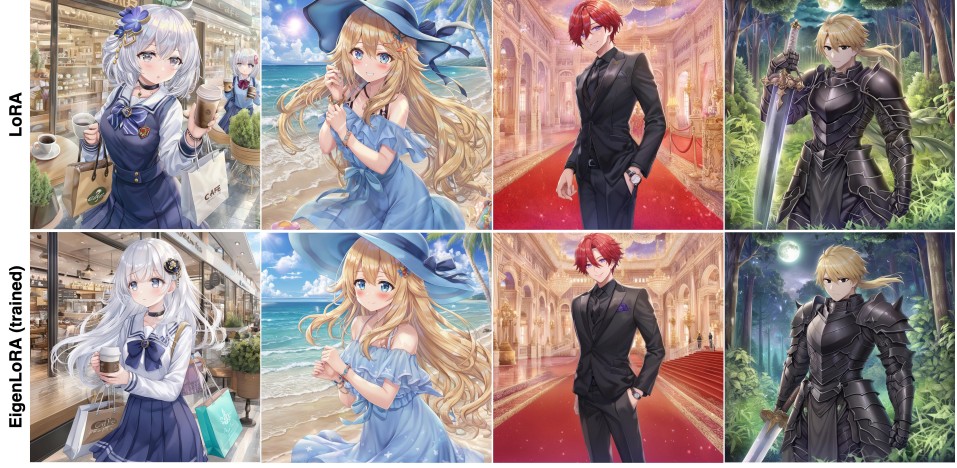

Figure 8: Comparison of generated images by LoRA and EigenLoRAx trained on Torino Aqua anime style images. For EigenLoRAx, we utilized 12 components with only trainable coefficients to finetune the base model.

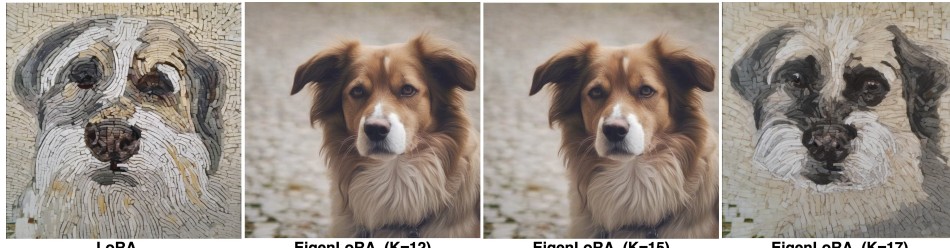

| LoRA | EigenLoRA (K=12) | EigenLoRA (K=15) | EigenLoRA (K=17) |

Figure 9: Failure Case: EigenLoRAx may fail if an important component is missing from the initialized subspace i.e. the shared subspace is incomplete, which may happen due to inadequacy in the number of initial adapters or due to the majority of the adapters being of bad quality. E.g., the model may have lost the essential "mosaic" property when generating an image for the prompt: "mosaic picture of a dog."

## A.5 ADDITIONAL EXPERIMENTS

Furthermore, we also performed a 3D object pose estimation (Angtian et al., 2021; Kaushik et al., 2024) finetuning experiment using a modified ResNet-101. The task of 3D object pose estimation involves the prediction of three rotation parameters (azimuth, elevation, in-plane rotation) of an object relative to the camera. The pose estimation error between the predicted rotation matrix and the ground truth rotation matrix is given as $\Delta(R_{pred}, R_{gt}) = \frac{|| \log_b(R_{pred}^{\mathsf{T}} R_{gt})||_F}{\sqrt{2}}$ We show the results for the $\frac{\pi}{6}$ accuracy threshold for this experiment.

Table 12: 3D object pose estimation accuracy ($\frac{\pi}{6}$ threshold)

| Method | Param | Airplane | Motorbike | Boat | Bottle | Bus | Car | Average |
|---|---|---|---|---|---|---|---|---|
| LoRA ($r = 16$) | 215K | 79.9 | 80.1 | 71.5 | 89.8 | 90.1 | 96.6 | 84.67 |
| VeRA ($r = 256$) | 40K | 68.4 | 72.4 | 64.3 | 88.4 | 87.2 | 94.4 | 79.18 |
| EigenLoRAx ($K = 2$) | 16K | 81.4 | 80.0 | 71.4 | 90 | 92.3 | 97.5 | 85.43 |

## B METHOD ANALYSIS AND ABLATION

Through a rigorous comparative analysis of EigenLoRAxs and their target LoRAs, we identified that the most pronounced reconstruction discrepancies manifest in the initial and terminal layers of the neural network, as depicted in Figure 10. Allowing the EigenLoRAx PCs in these layers to undergo fine-tuning along with the coefficients can alleviate failure scenarios, thereby alleviating the need for comprehensive model fine-tuning.

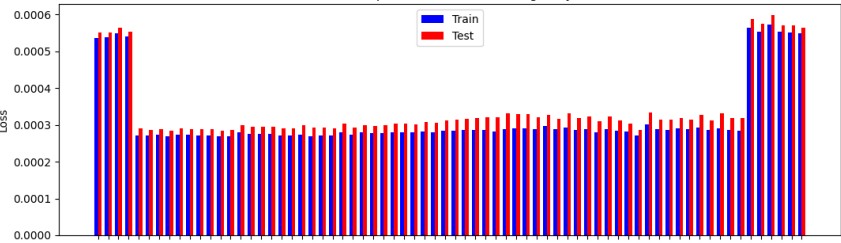

Figure 10: Average reconstruction error between EigenLoRAx and a set of LoRAs for all UNet layers in a stable diffusion model.

## B.1 How to Choose $K$ Principal Components and $r$ for EigenLoRAx

We perform an ablation study on the selection of EigenLoRAx principal components ($K$). Our analysis concentrates on one experiment as shown in Figure 13, specifically pertaining to the MRPC task within the GLUE (Wang et al., 2019) benchmark. The analysis in Figure 11 shows the training loss in relation to increasing number of EigenLoRAx principal components $K$, as well as the explained variance of the LoRAs used to initialize the EigenLoRAx in Figure 12. We find, empirically, that choosing EigenLoRAx PCs for the explained variance of $50 - 80\%$ of the LoRAs used to initialize EigenLoRAx is sufficient for a robust initialization. This is shown in Figure 12 where we choose $K = 8$ which roughly corresponds to the explained variance of $55 - 60\%$. We further ablate this choice in Figure 11, where although substantial improvements are evident up to $K = 8$, an increase in the number of $K$ thereafter yields only marginal gains, demonstrating diminishing returns as the number of components increases. The parameter $r$ in EigenLoRAx does not equate the *rank* parameter in LoRA and its variants. It reflects the dimensionality of the EigenLoRAx coefficients. Although $r = 1$ works well, we observe slight performance improvements as we increase this value as shown in Figure 14. Increasing this value corresponds to a small amount of parameter increase. We observe no finetuning instability by changing this value and recommend that it can be set to anywhere between 1 and the rank of the LoRAs used to initialize EigenLoRAx.

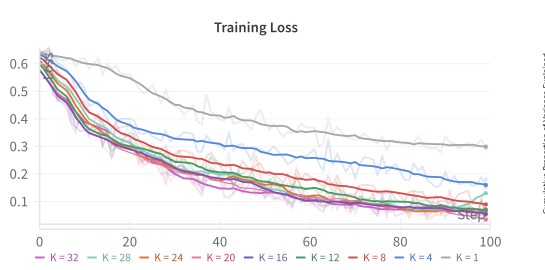

Figure 11: Training Loss convergence for different numbers of EigenLoRAx PCs

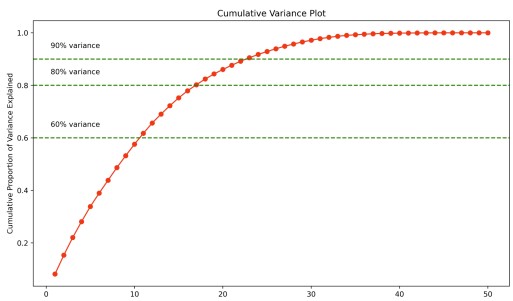

Figure 12: Explained Variance for increasing number of PCs

Figure 13: Ablation of Number of EigenLoRAx Principal Components

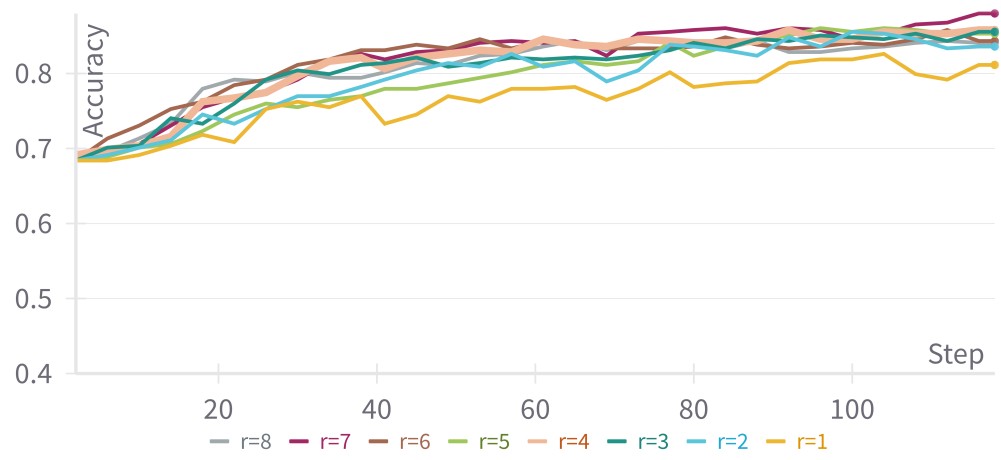

Figure 14: Ablation for the EigenLoRAx's $r$ hyperparameter. This experiment was done for the MRPC task in the GLUE benchmark.

## B.2 Failure Cases

Figure 9 illustrates a potential failure case of EigenLoRAx, where the incorrect number of principal components (PCs) was selected. In this instance, the "mosaic style" information was excluded from the principal subspace identified by EigenLoRAx due to an insufficient number of PCs. However, this issue can be resolved by selecting a larger number of PCs, as the extended principal subspace contains the necessary information for the task.

Another hypothetical failure scenario arises if the domain gap between the low-rank adapters used to initialize EigenLoRAx and the downstream task is significantly large. Although we do not observe such a case in our experiments, it is plausible that under such conditions, EigenLoRAx might underperform. This issue could potentially be mitigated by allowing only a subset of PCs to remain trainable, enabling the model to adapt more effectively to the target domain.

A further observed limitation of EigenLoRAx occurs in complex tasks like Text-to-Image generation, which may extend to other tasks as well. If the majority of LoRAs used to initialize EigenLoRAx encode biases (e.g., related to gender, race, or context), these biases tend to propagate into Eigen-LoRAx outputs. While such biases are a common issue in deep learning models trained using stochastic gradient descent or similar methods, addressing them remains a critical area of future work. We consider this an important avenue for improvement and discuss the broader implications in Appendix C.

## B.3 Impact of LoRA adapter quality on EigenLoRAx PC initialization

To evaluate EigenLoRAx's robustness to adapter quality and its resistance to noise, we conducted an ablation study on a subset of tasks of the NLU experiment specified in Section 4.2. Specifically, we generated EigenLoRAx adapters using LoRA matrices with varying levels of random noise added. The results are shown in Table 13

Table 13: EigenLoRAx performance on subset of GLUE task using noisy LoRA adapters for initialization

| Noise Level | CoLA | MRPC | RTE | STS-B | Avg |
|---|---|---|---|---|---|
| 5% | 60.51 | 85.45 | 74.73 | 89.9 | 77.65 |
| 15% | 57.53 | 83.09 | 72.92 | 89.9 | 75.86 |
| 30% | 55.23 | 76.47 | 71.84 | 89.8 | 73.34 |

The results show that EigenLoRAx exhibits only minor performance changes even as noise levels increase significantly, indicating some robustness to adapter quality. This suggests that EigenLoRAx can still perform effectively without high quality adapters. However, there is a limit to this robustness. If the signal-to-noise ratio (SNR) in the initial LoRA matrices becomes extremely low—where the LoRAs primarily encode noise rather than meaningful information—the effectiveness of EigenLoRAx diminishes. In such cases, the principal components (PCs) extracted by EigenLoRAx would correspond to random directions in the parameter space. Consequently, EigenLoRAx's performance would resemble that of random matrix methods, such as VeRA and NoLA. These methods rely on a large number of random components or bases to approximate meaningful results. While they can achieve reasonable performance, they require fine-tuning a substantially larger number of weights associated with these large number of random components, leading to less efficient learning compared to Eigen-LoRAx. This highlights an important consideration: for EigenLoRAx to maintain its efficiency and effectiveness, the initial LoRA matrices must contain at least a minimal level of meaningful signal. This requirement ensures that EigenLoRAx can leverage the structured information encoded in the LoRAs while avoiding the inefficiencies of purely random approaches.

## B.4 Forward pass and backward pass FLOPs

While it is obvious that EigenLoRAx utilized significantly less number of model parameters as the number of tasks in a domain increase, we show that even in terms of floating point operations on a single task, EigenLoRAx is more efficient than LoRA for our experiments. Even for a single task, the number of floating point operations or multiply-accumulate operations in a forward pass for

EigenLoRAx is lower than LoRA for all our experiments. Here are the comparisons of the floating point operations (FLOPs) for the forward (fwd FLOPs) and including backward pass (fwd+bwd FLOPs) for each of the Image Classification and GLUE benchmark (batch size = 1) (MFLOPs - MegaFlops):

Table 14: Floating Point Operation calculations for GLUE Benchmark experiment

| Method | Training Parameters | fwd FLOPs | fwd+bwd FLOPs |
|---|---|---|---|
| LoRA | 1.2M | 97,930 MFLOPS | 293,800 MFLOPS |
| VeRA | 25K | 106,390 MFLOPS | 319,170 MFLOPS |
| EigenLoRAx | 12K | 97,030 MFLOPS | 291,080 MFLOPS |

Table 15: Floating Point Operation calculations for Image Classification experiment

| Method | Training Parameters | fwd FLOPs | fwd+bwd FLOPs |
|---|---|---|---|
| LoRA | 36K | 33,773.8 MFLOPS | 101,322 MFLOPS |
| VeRA | 18K | 33,744.8 MFLOPS | 101,234 MFLOPS |
| EigenLoRAx | 96 | 33,730.2 MFLOPS | 101,191 MFLOPS |

## B.5 COMPARISON OF PARAMETER COUNT OF LORA AND EIGENLORAX

Table 16 shows the comparison of the number of trainable parameters for LoRA vs EigenLoRAx for RoBERTa$_{base}$. The values of $K$ for EigenLoRAx and $rank$ for LoRA range from 1 - 512. We can clearly observe that EigenLoRAx requires less parameters for values of $K$ like 32 and 64 than a LoRA with $rank$ = 1 requires.

Table 16: Parameter counts for LoRA and EigenLoRAx across different values of $K$ and the LoRA rank.

| Method | $K, rank = 1$ | $K, rank = 2$ | $K, rank = 4$ | $K, rank = 8$ | $K, rank = 16$ | $K, rank = 32$ | $K, rank = 64$ | $K, rank = 128$ | $K, rank = 256$ | $K, rank = 512$ |
|---|---|---|---|---|---|---|---|---|---|---|
| LoRA | 37K | 74K | 147K | 295K | 590K | 1.2M | 2.4M | 4.7M | 9.4M | 18.9M |
| EigenLoRAx ($r$=1) | 48 | 96 | 192 | 384 | 768 | 1.5K | 3K | 6K | 12K | 25K |
| EigenLoRAx ($r$=2) | 96 | 192 | 384 | 768 | 1.5K | 3K | 6K | 12K | 25K | 49K |
| EigenLoRAx ($r$=4) | 192 | 384 | 768 | 1.5K | 3K | 6K | 12K | 25K | 49K | 98K |
| EigenLoRAx ($r$=8) | 384 | 768 | 1.5K | 3K | 6K | 12K | 25K | 49K | 98K | 197K |
| EigenLoRAx ($r$=16) | 768 | 1.5K | 3K | 6K | 12K | 25K | 49K | 98K | 197K | 393K |
| EigenLoRAx ($r$=32) | 1.5K | 3K | 6K | 12K | 25K | 49K | 98K | 197K | 393K | 786K |

## B.6 TIME REQUIRED TO CALCULATE PRINCIPAL COMPONENTS

To calculate the Principal Components, EigenLoRAx uses either SVD or PCA. There is no significant effect on the performance based on the choice of the algorithm. The calculation of the PCs is extremely efficient and can be done naively on a CPU. Table 17 shows the timing calculation for 192 sets of LoRA weights (A+B matrices) using approximately 17k MB of memory in total.

Table 17: Runtime comparison of SVD (low rank, with different iterations) and PCA (full rank, no iterations) across varying component sizes.

| Method | 16 components | 32 components | 64 components | 128 components |
|---|---|---|---|---|
| SVD low rank (niter=10) | 50s | 70s | 94s | 168s |
| SVD low rank (niter=50) | 140s | 212s | 334s | 614s |
| PCA full rank | 430s | 610s | 762s | 940s |

## C  BROADER IMPACT AND IMPLICATIONS

This work presents a novel parameter-efficient method for deep learning methods utilizing open source, pretrained Low-Rank Adaptation (LoRA) models. By substantially reducing the computational and memory demands of training and inference, our approach creates a more sustainable and environmentally friendly deep learning paradigm. Our method democratizes accessibility to larger models, making them accessible to researchers and practitioners with limited resources. Furthermore, by harnessing pretrained models, our method can accelerate development and diminish the need for extensive data collection. However, we recognize the inherent risks associated with the use of pretrained models. These include potential biases (racial, gender, etc.), explicit content, since there is no guarantee of the data or method used in training the model, and the potential presence of malicious code. Appropriate caution is advised when using unverified, open-source models.

