# OpenReview forum: "EigenLoRAx: Efficient Low Rank Adaptation Using Recycled Principal Subspaces"
_ICLR.cc/2026/Conference — ICLR 2026 Conference Withdrawn Submission_

### Official Review · Reviewer_fRxi · 2025-10-29

**Soundness:** 3
**Presentation:** 2
**Contribution:** 3
**Rating:** 4
**Confidence:** 4

**Summary:**

This paper introduces EigenLoRAX, designed to recycle the vast and growing number of publicly available LoRA. Core contribution is to extract a shared, low-dimensional principal subspace from a series of existing adapters trained on related tasks. Adaptation to new tasks by learning only a small set of lightweight parameters for the fixed basis. This method has improvements in parameter efficiency, training speed, and memory footprint for inference, which is particularly relevant for on-device deployment scenarios.

**Strengths:**

Originality: The specific approach of extracting principal components from a pool of pretrained LoRA adapters and using them as a basis for new task adaptation is novel.

Quality: The experimental validation is comprehensive, spanning multiple modalities and diverse tasks. Results demonstrate consistent benefits in parameter efficiency and memory usage.

Clarity: The main method is clearly explained with helpful visualizations (Figure 1). Algorithm 1 provides a clear procedural description.

Significance: The work addresses important practical concerns about deployment efficiency and sustainability of large models. The demonstrated 100× parameter reduction and 18× memory efficiency improvements are significant for resource-constrained scenarios.

**Weaknesses:**

W1: Invalid Theoretical Foundation: The upper bound derived from Theorem A.1 is clearly invalid. The error originates in equation (6) in the proof process, where an upper bound was obtained for $R(h^{\mathcal{E}})- R(h^{\ast})$. However, in the derivation leading to equation (6), it is evident that $R(h^{\ast})-R(h^{\mathcal{E}})$ was transformed into the form of a lower bound, resulting in an erroneous proof process. Therefore, the final result has not been proven.

W2: Under-explored Connection to Related Fields: The claim of being "among the first to recycle pretrained adapters" overlooks substantial related work. Task Arithmetic methods [1] extensively explore combining multiple adapters through algebraic operations on weights. While EigenLoRAX's extraction differs technically, the fundamental goal of leveraging multiple adapters overlaps significantly. Additionally, the paper misses connections to the extensive literature on subspace methods in transfer learning and domain adaptation [2], which has long explored shared low-dimensional structures across different distributions. The paper should: (1) Explicitly distinguish subspace extraction from adapter merging/Task Arithmetic. (2) Provide empirical comparisons with Task Arithmetic baselines.

[1] Ilharco G, Ribeiro M T, Wortsman M, et al. Editing models with task arithmetic[J]. arXiv preprint arXiv:2212.04089, 2022.

[2] Asgarian A, Ashraf A B, Fleet D, et al. Subspace selection to suppress confounding source domain information in AAM transfer learning[C]//2017 IEEE International Joint Conference on Biometrics (IJCB). IEEE, 2017: 456-463.

**Questions:**

Q1: Regarding Theorem A.1: Given the identified logical flaw in the proof, can you provide a corrected proof for the generalization bound? If a correct proof is not available, how do you propose to revise the paper's theoretical claims to accurately reflect what has been demonstrated?

Q2: The method assumes that LoRAs used to build the subspace are from a related "task domain." Does EigenLoRAX still work when the LoRAs and the target task are completely unrelated?

If the authors can adequately address the above concerns, I would be willing to raise my score.

---

### Official Review · Reviewer_FmfA · 2025-10-29

**Soundness:** 4
**Presentation:** 4
**Contribution:** 4
**Rating:** 0
**Confidence:** 5

**Summary:**

The paper proposes EigenLoRAx, a principled way to recycle a bank of trained LoRA adapters by extracting a shared principal subspace and learning only task-specific combination coefficients with an optional orthogonal “augmentation” for very low-resource cases. Training is head-only; the backbone stays frozen. The idea is simple, well motivated, and broadly useful: retain multi-domain knowledge while cutting parameters, memory, and training time by large factors. Experiments across NLP and vision are solid and ablations are thoughtful. In short, technically this is a strong and well-written paper.

**Strengths:**

A clean, general idea with clear practical value. Compute a principal subspace from existing LoRA weights and adapt by learning coefficients, keeping the backbone intact. The paper reads crisply, the algorithm is transparent, and the experiments show competitive accuracy with far fewer trainable parameters and lower memory. Coverage across modalities and tasks increases confidence, and the analysis of low-resource augmentation is helpful. If judged purely on technical grounds, I would advocate for acceptance.

**Weaknesses:**

From a technical standpoint I don’t have blocking concerns. The approach assumes a reasonably rich adapter bank; the limits under strong domain shift could be characterized more fully, and some results would benefit from multi-seed statistics and stricter equal-budget comparisons against the very latest PEFT/MoE variants. These are refinements rather than flaws.

However, there is a policy issue: a version with the same core method and experiments appears to be already published in CVPR 2025 Workshops (archival proceedings). If that counts as prior publication under ICLR policy, the submission is ineligible and should be rejected on policy grounds.

**Questions:**

No technical questions for the authors. If the AC rules the submission eligible, please clarify in the camera-ready: (i) explicit differences vs. the CVPRW version; (ii) any new experiments or analyses unique to this submission.

**Details Of Ethics Concerns:**

A published version appears at CVPR 2025 Workshops (archival proceedings):
https://openaccess.thecvf.com/content/CVPR2025W/CV2/html/Kaushik_EigenLoRAx_Recycling_Adapters_to_Find_Principal_Subspaces_for_Resource-Efficient_Adaptation_CVPRW_2025_paper.html
My side-by-side reading indicates the same core method and experiments, with only minor framing/presentation differences. If CVPRW is considered archival, this likely violates ICLR’s prior-publication policy.

---

### Official Review · Reviewer_n6XQ · 2025-10-30

**Soundness:** 2
**Presentation:** 2
**Contribution:** 3
**Rating:** 6
**Confidence:** 4

**Summary:**

This paper introduces EigenLoRAx, a parameter-efficient fine-tuning (PEFT) method that aims to leverage existing pre-trained Low-Rank Adaptation (LoRA) adapters. The core idea is based on the hypothesis that LoRA adapters trained on related tasks share a common, low-dimensional principal subspace. EigenLoRAx extracts this shared subspace by performing Principal Component Analysis (PCA) or Singular Value Decomposition (SVD) on the weights (e.g., A or B matrices) of a set of existing LoRA adapters for a given base model. This results in a fixed set of $K$ principal components ($\mathcal{V}_K^T$) that represent task-invariant knowledge. For adapting to a new task, EigenLoRAx freezes these principal components and learns only a small set of task-specific coefficients ($\alpha$) that linearly combine the components. This drastically reduces the number of trainable parameters compared to standard LoRA. The method also proposes augmenting the subspace with orthogonalized random vectors in low-resource scenarios where few adapters are available. The paper claims significant reductions in parameters (up to 100x), faster convergence, and improved memory efficiency for inference, especially for serving multiple tasks. Experiments are shown across image classification, NLU (GLUE), and text-to-image generation.

**Strengths:**

1. Novelty: The central concept of "recycling" existing, publicly available LoRA adapters by extracting a shared subspace is highly novel and addresses a relevant issue of underutilized resources in the ML community. This proposes a fundamentally different approach to PEFT.

2. The core idea and the overall workflow (Figure 1 ) are presented clearly.

3. Potential for Extreme Efficiency: If the core assumption holds true, the method offers a pathway to drastically reduce fine-tuning parameters (learning only $K$ coefficients instead of full $r \times n$ matrices) and significantly decrease memory footprint during multi-task inference by sharing the principal components

**Weaknesses:**

1. Insufficient Evaluation of Performance Trade-offs: The paper emphasizes parameter reduction (e.g., "up to 100x" ) but does not adequately evaluate the potential performance degradation. Table 2  shows EigenLoRAx (12K params) matching LoRA (1.2M params), but this large LoRA rank (r=32) might be suboptimal for RoBERTa-base on GLUE. A fairer comparison would involve tuning LoRA rank to match EigenLoRAx's performance and then comparing parameters, or matching parameters and comparing performance. The zero-shot results in Table 4 show an average performance ratio of only 0.88 compared to LoRA, indicating a clear performance drop

2. Sensitivity and Practicality Concerns: The quality and effectiveness of the extracted subspace likely depend heavily on the number ($d$), quality, and diversity of the adapters used for its construction. How many adapters are needed? What if they are poorly trained or cover a narrow domain? How is the number of principal components ($K$) chosen, and how sensitive is the performance to $K$ and the initial adapter set? These crucial practical aspects are not sufficiently addressed. The proposed augmentation strategy for low-resource scenarios appears ad-hoc and lacks strong justification

**Questions:**

1. Can the authors provide stronger empirical evidence for the core assumption? For example, experiments showing performance on new tasks that are significantly different (e.g., cross-domain, cross-lingual) from the tasks used to build the subspace?

2. How sensitive is the method to the number ($d$) and selection of adapters used for subspace construction? What is the minimum $d$ required for good performance? How does performance degrade if the initial adapters are noisy or from a narrow domain?

**Details Of Ethics Concerns:**

Seems prepublished on 2025 CVPRW.

---

### Official Review · Reviewer_ZWKi · 2025-10-31

**Soundness:** 3
**Presentation:** 3
**Contribution:** 2
**Rating:** 6
**Confidence:** 4

**Summary:**

This paper proposes EigenLoRAx, a new PEFT method that leverages multiple fine-tuned LoRA adapters to construct a shared principal subspace for fast adaptation to new downstream tasks. By reusing existing adapters, EigenLoRAx reduces trainable parameters, accelerates convergence during training and improves memory efficiency at inference. Extensive experiments across vision, language, and text-to-image tasks demonstrate its versatility and effectiveness.

**Strengths:**

1. This paper addresses a practical and timely problem. Many existing LoRA adapters are underutilized for rapid adaptation.
2. The motivation is sound. Different downstream tasks may share overlapping low-rank subspaces, and viewing LoRA from a recycling perspective is novel and insightful.
3. The theoretical insights are solid and well presented. The question abstraction (in Section 3.1) and the theoretical formulation (in Section 3.2) are impressive.
4. The experimental evaluation is comprehensive, spanning image and text classification, instruction tuning, and image generation across diverse pre-trained models.

**Weaknesses:**

1. EigenLoRAx depends on having multiple fine-tuned LoRA adapters, which limits its general applicability.
2. The parameter reduction mainly arises from the fine-tuning structure itself. Eq. (1) appears conceptually similar to VeRA, weakening novelty.
3. Claims of faster convergence lack strong evidence. Although the paper provides a training loss curve on the CoLA dataset (Fig. 3), this single example and the absence of deeper analysis make it difficult to convincingly support the authors’ statement about convergence improvement.
4. While the experiments are extensive, their organization and consistency could be improved. The experimental setup is not entirely clear or uniform across sections, for instance:
a. Some experiments include PiSSA, others do not.
b. The choice of LoRA and VeRA rank values varies without sufficient explanation.
c. Table 1 omits the rank specification for VeRA.
These inconsistencies reduce the overall clarity and reproducibility of the experimental results.

**Questions:**

1. EigenLoRAx assumes the existence of multiple fine-tuned LoRA adapters (e.g., LoRA1 and LoRA2) and constructs shared subspaces VA and VB from them. For a new task, instead of training a full LoRA3, EigenLoRAx fine-tunes only small coefficient matrices alpha-B3 and alpha-A3. My understanding is that the total storage consists of the shared subspaces VA and VB and the task-specific coefficients alpha-Bi and alpha-Ai, for each task. If this interpretation is correct, does compressing the existing adapters into the shared subspace affect the performance of the original tasks (e.g., LoRA1 and LoRA2)?
In addition, the paper states in line 289 that the complexity is O(2Kl(d+n)). It would be helpful if the authors could elaborate on how this expression is derived.

2. In the Low-Resource Scenario (line 378), there is no comparison with baselines. The setup looks more like an ablation study. Could the authors clarify this?

3. The STS-B results appear unusual in Table 3 (e.g., –0.73 and 0.11). Could the authors specify which evaluation metric was used for STS-B and explain why such negative or near-zero values occur?

4. In the appendix, the random seed is reported as 42. Are the results in the main text based on a single random run, or are they averaged over multiple seeds? Clarifying this would help assess the robustness of the reported results.

Improvements:
It is recommended to carefully review the experimental section for clarity and consistency.
a. Please ensure that the rank and K values for each method are explicitly stated.
b. Consider standardizing the number of significant digits across all reported results.
c. In the Low-Resource Scenario section (line 378), the descriptions of RANDOM and +RAND in Table 3 are easily confused.

---

### Note · Authors · 2025-11-12

**Comment:**

We acknowledge that our work was previously accepted to a workshop, although it is unclear whether those proceedings are archival. To err on the side of caution, we have decided to withdraw our submission from ICLR.

**Withdrawal Confirmation:**

I have read and agree with the venue's withdrawal policy on behalf of myself and my co-authors.